# Weakly Supervised Pose Estimation of Surgical Instrument from a Single Endoscopic Image

**DOI:** 10.3390/s24113355

**Published:** 2024-05-23

**Authors:** Lihua Hu, Shida Feng, Bo Wang

**Affiliations:** 1College of Computer Sciences and Technology, Taiyuan University of Science and Technology, Taiyuan 030024, China; hlh@tyust.edu.cn (L.H.); s202120110672@stu.tyust.edu.cn (S.F.); 2State Key Laboratory of Multimodal Artificial Intelligence Systems, Institute of Automation, Chinese Academy of Sciences, Beijing 100190, China

**Keywords:** 6D pose estimation, weakly supervised learning, back-propagatable PnP

## Abstract

Instrument pose estimation is a key demand in computer-aided surgery, and its main challenges lie in two aspects: Firstly, the difficulty of obtaining stable corresponding image feature points due to the instruments’ high refraction and complicated background, and secondly, the lack of labeled pose data. This study aims to tackle the pose estimation problem of surgical instruments in the current endoscope system using a single endoscopic image. More specifically, a weakly supervised method based on the instrument’s image segmentation contour is proposed, with the effective assistance of synthesized endoscopic images. Our method consists of the following three modules: a segmentation module to automatically detect the instrument in the input image, followed by a point inference module to predict the image locations of the implicit feature points of the instrument, and a point back-propagatable Perspective-n-Point module to estimate the pose from the tentative 2D–3D corresponding points. To alleviate the over-reliance on point correspondence accuracy, the local errors of feature point matching and the global inconsistency of the corresponding contours are simultaneously minimized. Our proposed method is validated with both real and synthetic images in comparison with the current state-of-the-art methods.

## 1. Introduction

Robot-assisted minimally invasive surgery has become an important field of rapid development due to its enormous potential for application, where surgical instrument pose estimation is a key demand. However, instrument pose estimation faces two significant challenges: difficulty in obtaining stable feature points for surgical instruments due to their lack of texture and high reflectivity of surface materials, and the absence of labeled pose data for surgical instruments in monocular endoscopic images. Due to the two difficulties mentioned above, current surgical instrument pose estimation techniques often rely on additional positioning markers or control points, or the use of binoculars and depth cameras to provide annotation information [1,2]. For example, in order to enhance the active research of surgical tool identification and tracking, Lajkó et al. [3] utilize the surgeon’s visual feedback loop to compensate for robot surgery inaccuracies. Attanasio et al. [4] identify anatomical structures and provide contextual awareness in the surgical environment, while [5,6] improve robotic surgical precision with the help of recognition of surgical phases. Kitaguchi et al. [7] employed a convolutional neural network to achieve high-precision automatic recognition of surgical actions in laparoscopic images. Nguyen et al. [8] used a CNN-LSTM neural network model and an IMU sensor-based automatic evaluation system to classify and regress motion data in the JIGSAWS [9] dataset. Pan et al. [10] proposed an automatic surgical skill evaluation framework based on visual motion tracking and deep learning. That method introduced a Kernelized Correlation Filter (KCF) to capture key motion signals from the robot. In sum, although quite a number of different approaches are reported in the field, such works usually involve unnecessary assistant devices to the surgery, which cause inconvenience, even obstacles, to the surgery, and also hamper the system’s wide applicability to some extent.

Object pose estimation is a key issue in the field of computer vision. Deep learning-based object pose estimation methods are rapidly rising and have become the mainstream research direction because of their excellent adaptability. These methods are mainly divided into three categories: corresponding point-based methods, voting-based methods, and region-based methods. Here is a short review of such methods:

Corresponding point-based methods first define feature points on 3D models, then deduce the projection position of feature points [11,12] in the image through a regression network, and then calculate the posture by solving the Perspective-n-Point (PnP) problem for these feature points. Rad et al. [13] proposed the BB8 algorithm, which infers the image coordinates of eight bounding box corners of an object first and then estimates the pose of the object by solving the PnP algorithm. Similarly, the Yolo-6d method [14] borrowed the idea of You only look once (Yolo) [15] and used the Efficient Perspective-n-Point (EPnP) algorithm [16] to calculate the pose by predicting the projection of the eight corners of the target’s 3D bounding box on the 2D image. CullNet [17] improved Yolo-6d, using multi-channel output and confidence prediction to enhance pose estimation robustness. Other works [18,19] were proposed with improved results based on a similar idea. Hu et al. [20] proposed a segmentation-driven pose estimation algorithm. This algorithm first generates a foreground segment mask image, splits the image into several small patches, estimates object pose by every image patch containing the foreground, and finally fuses all the estimation results into a single final result. Pix2Pose [21] uses an encoder–decoder and generative adversarial network to infer projection coordinates of dense 3D model points and then estimate object pose with these corresponding 2D–3D point pairs. Similarly, Dense Pose Object Detector (DPOD) [22] uses UVmap to represent the pixel-level correspondence between 3D points and 2D projections and improves pose estimation accuracy by introducing a post-optimization module. Coordinates-based Disentangled Pose Network (CDPN) [23] also estimates the object pose based on pixel-level 2D–3D correspondence by separating translation and rotation components. Lu et al. [24] proposed a hybrid pseudo-labeling method that performs self-training by leveraging globally consistent pose estimates. Wang et al. [25] introduced self-supervised training into the field of pose estimation using SE(3) Equivariance networks.

The voting-based pose estimation method combines the advantages of sparse and dense correspondences, utilizing a voting mechanism to determine the object’s pose. Pixel-wise Voting Network (PVNet) [26] locates feature points through a voting mechanism, demonstrating strong occlusion robustness. Similarly, HybridPose [27] introduces new features into the voting method, improving pose estimation performance. He et al. [28] proposed the PVN-3D, leveraging the Hough voting scheme to robustly estimate object pose.

Region-based pose estimation methods, instead of using point matching pairs, utilize image region information to infer the object’s pose. Do et al. [29] proposed a Deep-6DPose algorithm. This method first utilizes Mask Region-based Convolutional Neural Network (R-CNN) [30] to obtain the foreground segmentation region, and then adds a pose regression branch to regress the object’s rotation matrix and translation vector. Liu et al. [31] proposed an indoor object pose estimation algorithm based on the modified Faster R-CNN [32] framework. Sundermeyer et al. [33] trained an auto-encoder to estimate the rotation matrix, first by taking the whole image as input, and then solved the translation vector separately. The Deep Iterative Matching (DeepIM) [34] algorithm uses a 3D model to render synthetic images at first, and then the original image, rendered image, and corresponding mask are inputted to the FlowNet network [35] to obtain decoupled translation and rotation vectors. Deng et al. [36] proposed a weakly supervised pose estimation method based on robot interaction. Wang et al. [37] proposed the Self6D algorithm, leveraging unlabeled Red Green Blue-Depth (RGB-D) data in a self-supervised manner by means of differentiable rendering.

Unlike other applications, the endoscopic image-based pose estimation of surgical instruments has special difficulties, as we mentioned at the very beginning of this work, which are, mainly, the lack of stable feature points of surgical instruments, the large reflection property, and the lack of labeled training data. To tackle such difficulties, we propose a weakly supervised neural network framework to estimate the pose from the surgical instrument contour in the endoscopic image and use the synthetical image contours and key feature points of the instrument to assist the network training. In particular, a 2D feature points inference module is proposed to predict the corresponding 2D feature points from the segmented instrument contour in the image, and a back-propagatable PnP network module [38] and a differentiable rendering module [39] are used to estimate the pose.

To train our network effectively, we employ a two-phase training strategy. We train the segmentation module, separated from other modules, with the help of TernausNet-16 [40]. Our focus is on training the other network modules after the segmentation network, which takes segmentation mask images as input and outputs pose estimation. In summary, the main contributions of this work are as follows: (1) We propose a weakly supervised pose estimation method for surgical instruments from single endoscopic images, assisted by synthesized endoscopic images of surgical instruments; (2) both local discrepancy errors of predicted feature points and the global inconsistency of the whole contour are taken into account in the training phase; (3) our proposed method outperforms many state-of-the-art methods with large real and synthetic data; and (4) we create a synthetic surgical instrument contour image dataset of about 50,000 samples with corresponding pose ground truth, providing data support for future research.

## 2. Materials and Methods

The method proposed in this paper is outlined in the following steps:

Firstly, we utilized an offline contour-based 3D reconstruction method to obtain 3D models of surgical instruments and selected 8 implicit feature points by the farthest point sampling method. Then, we imported these 3D models into Unreal Engine 5 (UE5) virtual engine to generate the pose labeled synthetic segmentation image training data.

Secondly, to effectively leverage the auto-generated training data, we utilized a surgical instrument image segmentation module to preprocess the real surgical scene image. This module employs a TernausNet-16 structure to infer the foreground mask image of surgical instruments from the real surgical scene images.

Finally, our instrument pose estimation network was trained by the auto-generated labeled data. The network consists of cascaded modules: a backbone module consisting of alternated cascaded channel–space tensor self-attention layers and multi-resolution convolution layers to infer the heat map style feature, an implicit points’ projection coordinates regression module, and a back-propagated PnP (BPnP) and differentiable rendering pose estimation module.

The overall framework of the proposed method in this paper is shown in Figure 1.

In the following section, the main modules will be elaborated.

### 2.1. Synthetic Training Data Generation

To address the stable feature tracking challenge caused by the lack of texture and specular reflection of surgical instrument surfaces, our method uses the foreground mask image of an instrument to infer the instrument’s pose. With the foreground mask image as an input of the pose estimation network, disturbances caused by the drastic change in image appearance can be eliminated, leading to more robust pose inference. We utilized UE 5 virtual engine to generate foreground mask image training data.

We used structure from the contour-based method to reconstruct the 3D models of three kinds of typical instruments: hooks, clippers, and suction, which are widely used in endoscopic minimally invasive surgery. After the 3D model reconstruction, eight implicit feature points were selected for each 3D model. Then, we imported the 3D models of instruments into UE 5 virtual engine, and we performed a uniform sampling within the reasonable range of the SE(3) pose space of the virtual camera. At each virtual camera pose sample point, a surgical instrument’s foreground mask image was generated by virtual engine rendering, with the projections of implicit feature points being calculated and sampled pose parameters being recorded. We augmented the training dataset by adding noise to the generated images. The comparison between a captured real surgical instrument image and a rendered image with normal material and lighting in UE 5 virtual engine is shown in Figure 2. The samples of generated training image data are shown in Figure 3.

We generated the training dataset of three types of instruments: hooks, clippers, and suction. The dataset contains 50,000 of samples, and each data sample includes a synthetic foreground mask image, its corresponding pose label, the implicit feature points’ 3D coordinates, and their 2D projections on the image. We utilized this dataset to train our model without further manual label effort, such that our model could be trained in a weakly supervised manner.

### 2.2. Instrument Segmentation Preprocessing

To better transfer the pose estimation network trained on synthetic data to estimate the pose of surgical instruments in real surgical scenes, we employed a surgical instrument segmentation network module before proceeding to our main pose estimation network. TernausNet-16 network, a popular one in the field of image segmentation, was used in this work. TernausNet-16 is an improvement and optimization based on the U-Net [41] architecture. Its model encoding layer replaces the original design with VGG16 [42], which features a deeper architecture that can capture richer and more abstract feature representations. This design gives TernausNet-16 stronger feature extraction and transfer learning capabilities, as VGG16 is typically pretrained on large-scale image classification datasets, allowing it to learn rich, general-purpose image features. Additionally, TernausNet-16 cleverly employs skip connections and deconvolution operations to enhance feature fusion between the encoder and decoder while retaining U-Net’s advantages in image segmentation. To further boost the model’s nonlinear capabilities, TernausNet-16 adopts batch normalization and Rectified Linear Unit (ReLU) activation functions, thereby improving the model’s expressiveness and generalization capabilities.

By utilizing foreground segmentation, we eliminated the interference caused by image appearance variations. We evaluated the performance of the segmentation module using the metrics Dice coefficient and Jaccard index to ensure that it effectively provided clear and accurate input for subsequent pose estimation tasks. Our experiments on real surgical scene data show that our method stably and robustly estimates the pose of surgical instruments, even in the presence of partial errors in the segmentation results.

In the main contour-based pose estimation network, we adopted a two-stage framework to estimate instrument pose from surgical instrument foreground mask images. At the first stage, our main framework estimated the 2D projection points of the implicit model feature points, and, at the second stage, the pose was predicted from the projection points and the contour points of the foreground mask.

### 2.3. Pose Estimation Module

#### 2.3.1. Multi-Scale Heatmap Feature Generation

The first part of our network is a feature extraction net, a multi-scale heatmap feature module, which consists of two sub-modules: a multi-resolution convolution module and an attention (CSTSA) net, inspired by a polarized self-attention network [43], as shown in Figure 4.

A multi-resolution convolution module is a four-stage cascaded convolution network that generates both high-resolution features and low-resolution features. The first stage consists of a high-resolution convolution branch, one-fourth of the original image resolution. Then, the following three stages gradually expand parallel branches, with the resolution of each new expanded branch being half of the lowest resolution of the previous stage. Then, the features of different resolutions are fused into a unified feature before being inputted to the next stage to preserve both the high-resolution local features and the low-resolution global features. After the last multi-resolution convolution stage, a feature map is generated by a Softmax layer.

To fully utilize the dependency information of different parts of the intermediate features after the third multi-resolution convolution stage, we leverage channel–space tensor self-attention network. Channel–space tensor self-attention network treats the intermediate features as a tensor and extracts attention from channel-U and channel-V attention and space only attention. Channel-U and channel-V attentions are the attentions acting on the space spanned by a channel direction vector and one base vector of space. The architecture of the CSTSA module is shown in Figure 5.

The two attention sub-modules extract the dependencies between space and channel, ensuring our model generates heatmap features that contain rich multi-scale information and fuse spatial and channel correlation information. We set the heatmap loss to supervise our heatmap as follows:(1)$$L_{h}=MSE(m,m^{*})$$

MSE⁡(·,·) is the mean squared error function in Equation (1), m is the estimated feature point heatmap, and $m^{*}$ is the ground truth feature point heatmap. With our synthetic dataset, as the ground truths of the projections of implicit feature points are known, the ground truth feature point heatmap is generated by setting impulses at all the points’ ground truth positions and convolving them with a Gaussian kernel with the following parameter setting:  σ = 8 pixels.

#### 2.3.2. Implicit Feature Point Projection Regression

After the extraction of feature heatmaps, we utilized a simple implicit feature point projection regression module to predict the coordinates. This module is a differentiable network to regress the pixel coordinates from the heatmap, and its architecture is shown in Figure 6.

#### 2.3.3. Surgical Instrument Pose Estimation

After the inference of the projected implicit feature points, we utilized the back-propagatable PnP(BPnP) to estimate the pose of the surgical instrument by solving the originally non-differentiable implicit function through gradient-based back-propagation. The BPnP loss function is the sum of the reprojection errors of the implicit feature points, as shown in Equation (2).
(2)$$\begin{array}{l} l_{p}(x,x^{*},y,z,K)=\sum_{i=1}^{n}{\left\|r_{i}\right\|}_{2}^{2}+\lambda {\left\|{r_{i}}^{*}\right\|}_{2}^{2}\\ =\sum_{i=1}^{n}{\left\|x_{i}-(y,z_{i},K)\right\|}_{2}^{2}+\lambda \sum_{i=1}^{n}{\left\|{x_{i}}^{*}-(y,z_{i},K)\right\|}_{2}^{2} \end{array}$$
where x is the inferred projections of implicit feature points, $x^{*}$ is the ground truth of the projections, y is the pose parameters, z is the 3D implicit feature points, K is the intrinsic matrix of the camera, $\;r_{i}$ is the reprojection error of the implicit feature point, and  π is the projection function of a 3D point to the image one.

We leverage the zero-point theorem to convert the task of solving pose parameters by minimizing the reprojection loss into solving the explicit equation derived from partial differentials of the reprojection loss with respect to the pose parameters, as shown in Equation (3).
(3)$$f(x,y,z,K)=\frac{\partial l_{p}(x,y,z,K)}{\partial y}=0$$

According to the implicit function theorem, the partial differential of the pose parameters with respect to the reprojection errors of the implicit feature point coordinates can be computed as follows:(4)$$\frac{\partial y}{\partial x}=-{\left[\frac{\partial f}{\partial y}\right]}^{-1}[\frac{\partial f}{\partial x}]$$

Thus, the explicit expression of the gradient of reprojection loss with respect to the network parameters θ is as follows:(5)$$\frac{dl_{p}}{d\theta }=\frac{\partial l_{p}}{\partial y}\frac{\partial y}{\partial x}\frac{\partial x}{\partial \theta }+\frac{\partial l_{p}}{\partial x}\frac{\partial x}{\partial \theta }$$

By minimizing the final loss function, this gradient is back-propagated in end-to-end to supervise the network parameters of the multi-scale heatmap feature module and the implicit feature point projection regression module, and finally, the surgical instrument pose is estimated.

To fully use the mask contour information, we also set a differentiable loss function to enhance the contour global consistency; the foreground rendering loss is defined as follows:(6)$$L_{r}=MSE(m_{r},m_{s})$$
where $m_{r}$ is the rendered foreground image, and $m_{s}$ is the segmented foreground image. We set the final loss as the weighted sum of the reprojection loss, the foreground consistency loss, and the heatmap loss, as follows:(7)$$l_{final}=l_{h}+\alpha l_{p}+\beta l_{r}$$

By minimizing this final loss, we trained our surgical instrument pose estimation in a weakly supervised manner. Our model utilizes synthetic foreground mask image data generated by a virtual engine as the training set and tackles the problems of stable tracking and lack of pose annotation in weakly supervised learning. In a real-data experiment, our method must be validated to be robust in estimating the surgical instrument pose, even for instruments with slightly different 3D shapes. In Equation (7), the hyper parameter α is set to 0.00005, and β is set to 1.2.

## 3. Results

To verify the validity of our method, experiments on both real surgery scenes and synthetic data were carried out. For real surgery scene experiments, we tested our method on our self-collected minimally invasive lung lobe resection endoscopic surgery dataset in collaboration with the Renmin Hospital affiliated with Peking University. For the synthetic data experiments, we tested our method on the UE 5 virtual engine generated instrument data. We undertook comparisons between the results of our method and the current state-of-the-art single-view Red Green Blue (RGB) image-based object pose estimation techniques on various popular metrics. To evaluate the impact of each submodule on our method, we conducted an ablation study on both real surgery scene data and synthetic data.

### 3.1. Metrics

The commonly used metrics of Average Distance of 3D points (ADD)(S)(0.1 d) [44,45], 2D projection error(5 px) [46], and 2D projection error(3 px) were used for the synthetic data experiments. The intersection over union (IOU) was used for real-scene data experiments. ADD refers to the average distance between the 3D model points obtained using ground truth poses and predicted poses. Typically, if the average distance of model points is less than 10% of the model diameter, the pose estimation is considered correct. ADD-S is a specific variant used for measuring symmetric objects. The 2D projection error refers to the average distance between the 2D feature points obtained from the ground truth pose projection and that from the predicted pose projection. The definitions of these metrics are shown as Equations (8)–(11).
(8)$$ADD=\frac{1}{n}\sum_{i=1}^{n}\left\|\left(Rx_{i}+t)-(\bar{R}x_{i}+\bar{t}\right)\right\|$$
(9)$$\;\;ADD-S=\frac{1}{n}\sum_{i=1}^{n}\underset{x_{j}}{\min}\left\|\left(Rx_{i}+t)-(\bar{R}x_{j}+\bar{t}\right)\right\|$$
(10)$$2D\;\mathrm{projection}\;\mathrm{error}=\frac{1}{m}\sum_{i=1}^{m}\left\|K(Rx_{i}+t)-K(\bar{R}x_{i}+\bar{t})\right\|$$
(11)$$IOU=\frac{\left|A\cap B\right|}{\left|A\cup B\right|}$$
where $x_{i}$ is the three-dimensional model point of the object, R is the predicted rotation matrix, t is the predicted translation vector, K is the camera intrinsic matrix, $\bar{R}$ is the ground truth of the rotation matrix, $\bar{t}$ is the ground truth of the translation vector, and $x_{j}$ is the nearest distance point in the corresponding points of the symmetric object. IOU:A and B represent the foreground masks of instrument reprojection and segmentation. 

### 3.2. Implementation Detail

We scaled the input image to a 640 × 480 resolution for both training and testing, as in [47]. We trained the network for 130 epochs. The initial learning rate was set to 2 × 10^5^. After 50 epochs, it was adjusted to 5 × 10^6^. After 100 epochs, it was further adjusted to 2 × 10^6^. After 120 epochs, it was set to 2 × 10^7^. To accelerate iterative training, the first 50 epochs were trained using only heatmap loss. The following 70 epochs were trained using a combination of heatmap loss and the reprojection error of the implicit feature points. The final 10 epochs were trained using the final loss. We used Adam as our optimizer with a momentum of 0.9 and a weight decay of 1 × 10^3^.

The experimental results are reported as follows:

### 3.3. Comparisons of Pose Estimation Results

#### 3.3.1. Real Surgery Scene Data Experiments

In our self-collected lung endoscopic surgery dataset, including 2000 image frames extracted from real surgical video recordings, we predicted the pose of three kinds of instruments: hook, suction, and clippers, using our method trained on our synthetic dataset of 50,000 pose labeled images. Figure 7 illustrates some examples of the reprojected masks of the three kinds of instruments under the pose inferred by our method. In Figure 7, the blue points on the mask are the estimated feature points, and these images show that the reprojected masks are closely overlaid on the instrument, which indicates that the estimated poses are accurate.

As pose ground truth is unavailable for real-scene data, we compared our method with state-of-the-art methods on the IOU metric. The comparison methods include the following: Segmentation-driven-pose [20], Transpose + RansacPnP [48], Higherhrnet + RansacPnP [49], and PVNet [26]; the results are shown in Table 1. The results show that our method outperforms the others by a significant margin.

#### 3.3.2. Synthetic Data Experiments

In our synthetic dataset experiments, we compared our methods with those methods on ADD(S)(0.1 d), 2D projection error(5 px), and 2D projection error(3 px) as the synthetic images contain the real pose information. Our dataset contains 50,000 images of 3 kinds of instruments in different poses (roughly the same number of images for each instrument). We train the compared models with 90% of the data from the dataset and the test models with the remaining 10% of the data. The results of the comparisons are shown in Table 2. The results show that the pose estimated by our method is significantly more accurate on all the three metrics.

### 3.4. Ablation Study

In order to investigate the effects of each module of our method, we conducted an ablation study with both real-scene and synthetic data. We employed various combinations of multi-resolution convolution (MrC) modules, channel–space tensor self-attention (CSTSA) modules, BPnP modules, and differentiable rendering pose estimation (DRPE) modules to assess the impact of different modules, and the results are shown in Table 3 and Table 4.

The above ablation study shows that each module has a positive impact on the results. More specifically, the DRPE module has made the most contribution to the IOU metric in real experiments. For synthetic data, all our modules have positive contributions to the ADD(S) metric, and DRPE and CSTSA have the greatest impact on the 2D projection metric.

## 4. Conclusions and Discussion

In this work, we have proposed a new weakly supervised method for 6D pose estimation of surgical instruments from single endoscopic images. Our method requires only virtual engine generated synthetic instrument foreground masks as training data, without extensive manually labeled effort. Our method has successfully addressed the common challenges of stable feature tracking and the lack of labeled data in real surgical instrument pose estimation. Experiments on both self-collected real surgical instrument datasets and synthetic data demonstrate that, compared to state-of-the-art methods, our method exhibits enhanced robustness and reliability in pose estimation tasks. Our work offers a new solution for surgical instrument pose estimation that is applicable not only to surgical instrument pose estimation tasks but also provides valuable insights for weakly supervised object pose estimation problems in other domains.

Since our method uses mask images for pose estimation, our method is less affected by the illumination changes and lack of salient textures on surgical instruments. However, the contour-based methods also have their inherent weaknesses. For example, how do you choose the implicit feature points on the instrument so that they are visible under most endoscope poses? If some of the model points are invisible, whether a visible subset of feature points could still reliably predict the pose, etc.

During the experiments, we observed that in some cases, only five or six feature points out of the total eight were visible, and their corresponding estimated poses were still accurate enough. We thought that this was largely aided by our global contour consistency enforcement since the contour shape itself already imposes some effective constraints on the possible pose space.

In the future, we would like to investigate the reliability of our method on other surgical instruments, as any real surgical operation usually involves a variety of tools. In addition, the current work is based on a single endoscopic image, in real surgery, video sequences are available, and the time domain continuity will be explored to further boost the pose estimation’s robustness and accuracy.

## Figures and Tables

**Figure 1 sensors-24-03355-f001:**
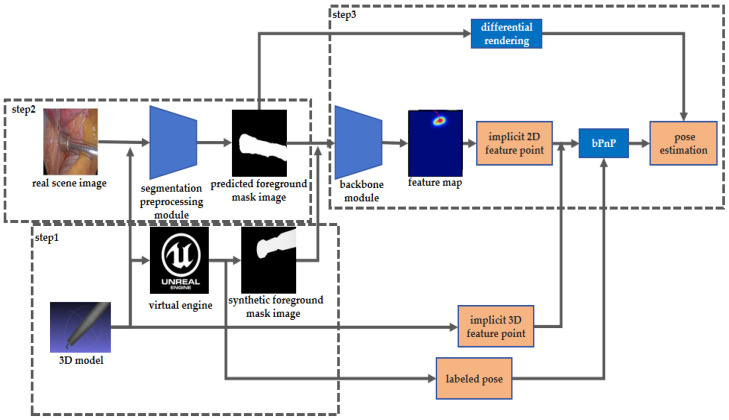
The overall framework of our weakly supervised surgical tools instrument pose estimation method. Three steps are involved. Step 1: Synthetic data generation. Step 2: Surgical instruments segmentation of real scenes. Step 3: Segmentation based surgical instruments pose estimation.

**Figure 2 sensors-24-03355-f002:**
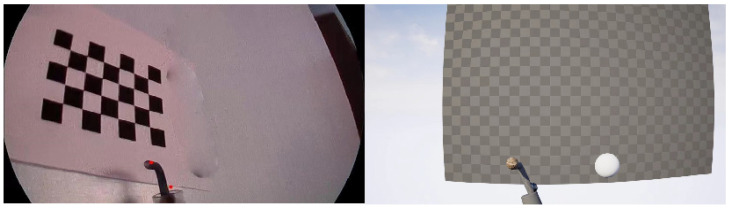
Comparison between a captured real surgical tool image and a rendered image with normal material and lighting in UE 5 virtual engine.

**Figure 3 sensors-24-03355-f003:**
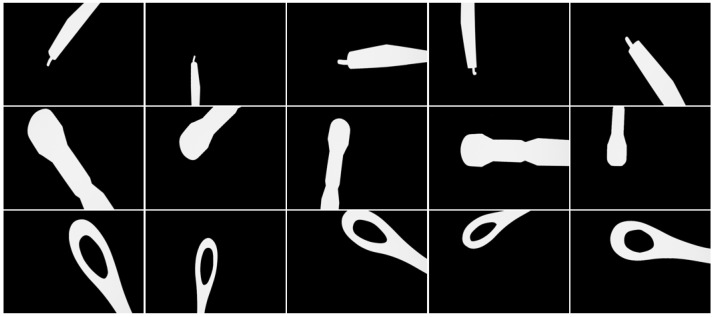
Auto-generated training dataset samples.

**Figure 4 sensors-24-03355-f004:**
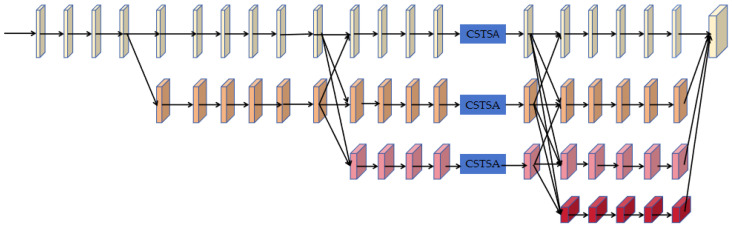
The multi-scale heatmap feature module to generate the feature map. The colors of the blocks indicate the feature map size. The yellow feature map is 1/4 the size of the original image, with 32 channels. The orange feature is 1/8 the size of the original image, with 64 channels. The pink feature is 1/16 the size of the original image, with 128 channels. The red feature is 1/32 the size of the original image, with 256 channels.

**Figure 5 sensors-24-03355-f005:**
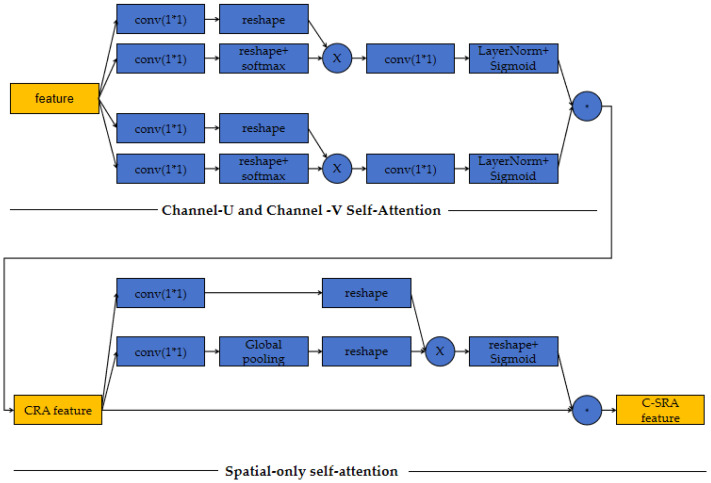
Architecture of the CSTSA module.

**Figure 6 sensors-24-03355-f006:**
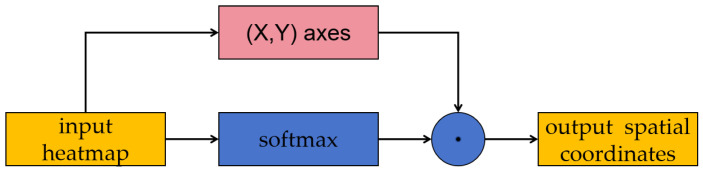
Architecture of implicit feature point projection regression module.

**Figure 7 sensors-24-03355-f007:**
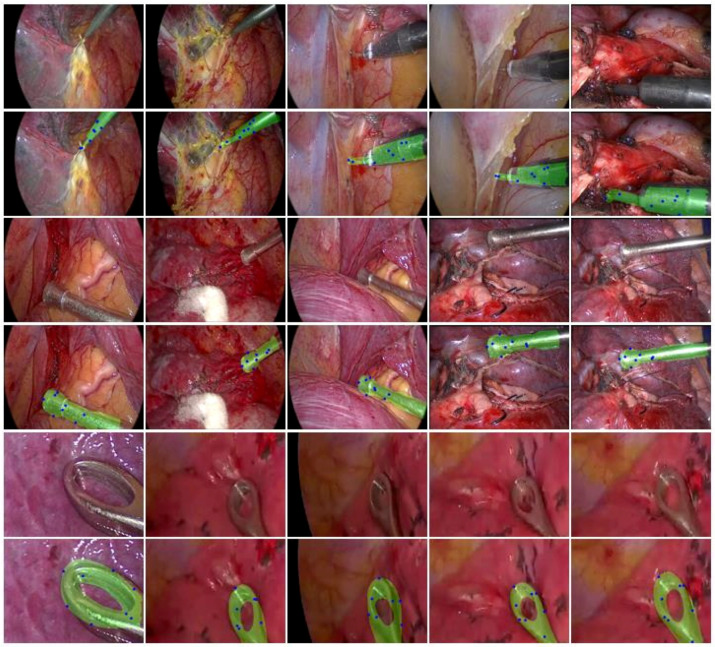
Examples of the reprojected masks of hook, suction, and clippers under the pose inferred by our method.

**Table 1 sensors-24-03355-t001:** Comparison of our method with state-of-the-art methods on the IOU metric.

Model	IOU			
	Sucker	Hook	Clipper	Average
Segmentation- driven-pose	48.20	42.91	26.42	39.18
Transpose + RansacPnP	41.22	43.68	41.58	42.16
Higherhrnet + RansacPnP	64.65	42.06	51.84	52.85
PVNet	59.24	33.67	45.59	46.17
Ours	67.06	57.71	52.89	59.22

**Table 2 sensors-24-03355-t002:** Comparison of our method with state-of-the-art methods, on the ADD(S)(0.1 d), 2D projection(5 px), and (3 px) metrics.

Model	ADD(S)(0.1 d)				2D Projection Error(5 px)				2D Projection Error(3 px)			
	Sucker	Hook	Clipper	Average	Sucker	Hook	Clipper	Average	Sucker	Hook	Clipper	Average
Segmentation- driven-pose	98.48	63.63	98.48	86.86	28.79	42.42	10.61	27.27	7.58	16.67	4.55	9.60
Transpose + RansacPnP	48.48	48.48	75.76	57.57	6.06	53.03	43.94	34.34	1.52	13.64	12.12	9.09
Higherhrnet + RansacPnP	100	77.27	100	92.42	37.88	86.36	69.70	64.65	10.61	56.06	34.85	33.84
PVNet	98.48	33.33	100	77.27	37.88	28.79	80.30	48.99	12.12	10.61	42.42	21.72
Ours	100	**86.36**	100	95.45	46.97	89.39	84.85	73.74	19.70	68.18	43.94	43.94

**Table 3 sensors-24-03355-t003:** Comparison of different combinations of our modules on the ADD(S)(0.1 d), 2D projection(5 px), and (3 px) metrics, with synthetic data.

Synthetic Data												
Model	ADD(S)(0.1 d)				2D Projection Error(5 px)				2D Projection Error(3 px)			
	Sucker	Hook	Clipper	Average	Sucker	Hook	Clipper	Average	Sucker	Hook	Clipper	Average
MrC	98.48	82.83	99.49	93.6	33.84	86.36	79.29	66.50	10.61	56.06	33.84	33.50
MrC + CSTSA	**100**	79.80	**100**	93.26	43.43	86.87	78.28	69.53	16.67	63.64	38.89	39.73
MrC + CSTSA + BPnP	99.49	86.36	100	95.28	44.44	86.36	81.82	70.87	17.68	65.15	44.44	42.42
MrC + CSTSA + BPnP + DRPE	100	86.36	100	95.45	46.97	89.39	84.85	73.74	19.70	68.18	43.94	43.94

**Table 4 sensors-24-03355-t004:** Comparison of different combinations of our modules on the IOU metric with real data.

Model	IOU			
	Sucker	Hook	Clipper	Average
MrC	61.32	41.02	31.31	44.55
MrC + CSTSA	63.56	46.09	30.17	46.61
MrC + CSTSA + BPnP	64.78	44.99	33.94	47.90
MrC + CSTSA + BPnP + DRPE	67.06	57.71	52.89	59.22

## Data Availability

Our synthetic surgical instrument foreground mask training dataset and real surgical scene test dataset will be shared on our website if the paper is accepted.

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
