# Peer review of "Weakly Supervised Pose Estimation of Surgical Instrument from a Single Endoscopic Image"

_sensors, 2024, doi:10.3390/s24113355_

Round 1
Reviewer 1 Report
Comments and Suggestions for Authors
The authors present a new deep learning-based model for estimating the pose of surgical tools from endoscopic images. This is a very relevant topic for the advancement of automation technologies in non-invasive surgery. However, as the authors mention, the difficulty of the problem is composed with a lack of labeled datasets.
To solve this problem, the authors propose a model based on several processes: an initial segmentation followed by key point identification and pose estimation using a differentiable PnP algorithm. The architecture is presented with enough detail. The lack of labeled data is solved using synthetic data generated using the Unreal 5 game engine.
The structure of the paper is OK. The introduction presents a good analysis of the current state of the art. An analysis describing how existing models fail in the setting proposed by the authors would be, however, welcomed. The abstract and initial paragraphs of the introduction should be revised as there are some sentences that are difficult to understand.
Regarding the content, as mentioned above, the model is presented in detail. However, there are some points that could be improved. For example, in equation (5) functions g and h have not been presented and their meaning is not obvious from the text.
The results use the typical metrics for 6D Pose estimation. The results presented in real-world conditions in Figure 7 are of very good quality. There is, however, a formatting problem with tables 1 and 3, that should be solved prior to publication.
Comments on the Quality of English Language
The quality of English is in general OK. The abstract and initial paragraphs of the introduction should be revised, as there are some complex sentences that could be improved for better readability.
Reviewer 2 Report
Comments and Suggestions for Authors
This paper presented a weakly-supervised pose estimation of surgical instrument from a single endoscopic image. The paper is easy to follow.
The topic of the research is relevant and current, yet the manuscript would definitely benefit from a more generic overview of the field of AI for Minimally Invasive Surgery, especially regarding the usability and real value of pose estimation. This may have a crucial input to skill assessment and workflow safety as well. Some ill-posed expressions, such as "Intelligent medical treatment" shall be replaced with more technical terms. See some adequate nomenclature e.g. in doi: 10.1109/JPROC.2022.3180350 or doi.org/10.1146/annurev-control-062420-090543
It would be beneficial for the research community especially to increase the replicability of the research, and particularly test the algorithm on well established databases, such as the JIGSAWS (see e.g., DOI: 10.12700/APH.20.8.2023.8.8). Many relevant approaches have been developed not fully covered in the Introduction part. (E.g,https://doi.org/10.1007/s00464-023-10335-z; doi: 10.3390/s23094496)
Minor issues:
A lot of the references lack a space before them. For example, within the 46th line of the Introduction section, the reference is cited like this: "feature points[24, 25]". This should be written like: feature points [24, 25].
The figure reference is inconsistent, sometimes it is referred to as Fig., but other times as Figure. It is recommended to use one of the two types, and use that for all of the figure references in the text.
It is recommended to avoid the use of bullet points to denote subsections within the pose estimation module. Instead, employing bold text would serve as a more effective way of indicating a subsection or use a second subsection indicator like in the third section (3.3.1.).
To improve the readability all of the equations should be centered.
In the metrics section the used metrics lack a more detailed explanation about how the metric works, for example what are the differences between the ADD and ADD-S. Why are both of these calculated?
It is advisable to reconsider the position and the format of the first three tables in the manuscript. First of all space after the first table is necessary, so that the 3.3.2. Synthetic Data Experiments section is not next to the table. It is hard to decide which column belongs to the main columns (ADD, 2D) in Table 2 and 3.To enhance readability, the inclusion of a vertical line delineating these main columns would be advantageous.
Comments on the Quality of English Language
Small typos:
● Introduction:
○ 37: [1,2] → [1, 2]
○ 47: Perspective-n-Point(PnP) → Perspective-n-Point (PnP)
○ 52: yolo → Yolo
○ 55: Hu et al[7]. → Hu et al. [7]
○ 90: we proposes → we propose
○ 102: images , assisted → images, assisted
○ 103: endoscopic images (multiple spaces between the words)
○ 108: truth, providing (multiple spaces between the words)
● Materials and methods:
○ 111: obtain3D → obtain 3D
○ 113: into Unreal (multiple spaces between the words)
● Pose estimation module:
○ 192: channel-space tensor self-attention(CSTSA) → attention (CSTSA)
○ Figure 5:
■ Sgmoid → Sigmoid
■ Golbal → Global
○ 221: function,in → function, in
● Surgical instrument pose estimation:
○ 267: with a some different 3D shapes → with slightly different 3D shapes
○ Eq(7),the → Eq(7), the
● Metrics:
○ 279: 3.1 → 3.1.
○ 282: intersection over union(IOU) → union (IOU)
○ 289: For IOU :A → For IOU: A
● Implementation Detail:
○ 292: 3.2 → 3.2.
● Real Surgery Scene Data Experiments:
○ 304: image frames (multiple spaces between the words)
● Synthetic Data Experiments:
○ 335: ( roughly → (roughly
○ 338: all the three metrics → all the three metrics.
● Ablation Study:
○ 345: tion(CSTSA) → tion (CSTSA)
○ 345: estimation(DRPE) → estimation (DRPE)
○ Table 3: Syntheticdata → Synthetic data
Abstract:
● 20: The meaning of the PNP abbreviation should be cited at the first mention.
Introduction:
● 50: Yolo abbreviation was used without explaining the meaning.
● 51: EPnP abbreviation was used without explaining the meaning.
● 61: DPOD abbreviation was used without explaining the meaning.
● 63: CDPN abbreviation was used without explaining the meaning.
● 69: PVNET abbreviation was used without explaining the meaning.
● 76: R-CNN abbreviation was used without explaining the meaning.
● 81: DeepIM abbreviation was used without explaining the meaning.
Materials and Methods:
● Figure 1:
○ Figure 1 title should contain more information about the picture (steps explanation). The text on the figure is hard to read due to their small size and the fact that a lot of the text is covered by a dashed line.
Synthetic Training Data Generation:
● 135: UE abbreviation was used without explaining the meaning.
Instruments Segmentation Preprocessing:
● This section should be on the following page.
● 175: ReLU abbreviation was used without explaining the meaning.
Pose estimation module:
● Figure 4:
○ 195: It is advisable to explain the different colors in the title.
● Figure 5:
○ Channel-U and Channel-V Self Attention and Spatial-only self attention texts are hardly seeable.
Implicit feature point projection regression:
● There should be a space between this subsection and the previous subsection's last sentence.
Results:
● 276: RGB abbreviation was used without explaining the meaning.
Real Surgery Scene Data Experiments:
● Figure 7:
○ The choice of the color for representing the estimated feature points is not great, because the red circle blends into the pictures.
Ablation Study:
● There should be a space between this subsection and the previous subsection.
● Table 3 should be on the following page.
This could become a fair paper, yet some wider introduction would make it more reader-friendly.
